# Analysis of E.U. Rapid Alert System (RASFF) Notifications for Aflatoxins in Exported U.S. Food and Feed Products for 2010–2019

**DOI:** 10.3390/toxins13020090

**Published:** 2021-01-26

**Authors:** Ahmad Alshannaq, Jae-Hyuk Yu

**Affiliations:** Department of Bacteriology, University of Wisconsin-Madison, 1550 Linden Drive, Madison, WI 53706, USA; alshannaq@wisc.edu

**Keywords:** RASFF notifications, mycotoxin, aflatoxin contamination, United States nuts, pistachios, border rejection

## Abstract

The most common, toxic, and carcinogenic mycotoxins found in human food and animal feed are the aflatoxins (AFs). The United States is a leading exporter of various nuts, with a marketing value of $9.1 billion in 2019; the European Union countries are the major importers of U.S. nuts. In the past few years, border rejections and notifications for U.S. tree nuts and peanuts exported to the E.U. countries have increased due to AF contamination. In this work, we analyzed notifications from the “Rapid Alert System for Food and Feed (RASFF)” on U.S. food and feed products contaminated with mycotoxins, primarily AFs, for the 10-year period 2010–2019. Almost 95% of U.S. mycotoxin RASFF notifications were reported for foods and only 5% for feeds. We found that 98.9% of the U.S. food notifications on mycotoxins were due to the AF contamination in almond, peanut, and pistachio nuts. Over half of these notifications (57.9%) were due to total AF levels greater than the FDA action level in food of 20 ng g^−1^. The Netherlands issued 27% of the AF notifications for U.S. nuts. Border rejection was reported for more than 78% of AF notifications in U.S. nuts. All U.S. feed notifications on mycotoxins occurred due to the AF contamination. Our research contributes to better understanding the main reasons behind RASFF mycotoxins notifications of U.S. food and feed products destined to E.U. countries. Furthermore, we speculate possible causes of this problem and provide a potential solution that could minimize the number of notifications for U.S. agricultural export market.

## 1. Introduction

Mycotoxins are unavoidable and unpredictable toxic fungal secondary metabolites produced by three major genera of soil-borne molds: *Aspergilli*, *Fusarium*, and *Penicillium* [1,2]. The most common, toxic, and carcinogenic mycotoxin found in human food and animal feed are the aflatoxins (AFs), especially aflatoxin B1 (AFB1) [3,4,5]. AFs have been reported to be present in a wide variety of crops, including corn, wheat, soy, rice, cottonseed, tree nuts, oilseeds, herbs, and spices. Animal byproducts such as milk, meat, and egg can also be at risk of AF contamination [1,6,7,8,9]. Moreover, AF contamination is one of the key foodborne risks that is greatly influenced by climate conditions. High temperatures, humid weather, and drought stress are favorable conditions for dissemination of and infestation by the primary AF-producing fungus *Aspergillus flavus* [10,11]. Therefore, the ongoing global warming is expected to elevate the levels of AF contamination, especially in fields at the pre-harvest stage [11,12,13,14,15].

The United States is continuing to be a major producer and exporter of tree nuts such as almonds, pistachios, walnuts, pecans, and hazelnuts, and is one of the leading exporters of groundnuts (peanuts) worldwide Table 1 and Table 2 [16]. In 2019, the market value of the U.S. tree nut and peanut exports to the world was $9.1 billion and $675 million, respectively. The European Union countries are the largest market for U.S. tree nuts, importing more than a third of all exported U.S. tree nuts, and they represent the third greatest market for exported U.S. peanuts [17,18,19,20]. Unfortunately, all these edible nuts are prone to fungal infestation and contamination with mycotoxins, especially AFs, that hamper the flow of the nuts across borders. Worryingly, increasing numbers of incidents where U.S. tree nuts exported to the E.U. countries have been rejected at the border because of AF contamination have occurred in the past decade.

Mycotoxin contamination results in more notifications than any other hazard in the Rapid Alert System for Food and Feed (RASFF), and the foremost toxin that has been associated with the notifications is AFs, especially in the nuts and nut products [1,21,22,23,24]. RASFF was established in 1979 by the E.U. countries to allow swift exchange of information on hazards in food and feed (chemical, biological, physical, and allergens) among the E.U. countries. All food and feedstuff imported into E.U. are checked by the competent authorities of the Member States, and when risks to public health are detected during these checks, information is disseminated through the RASFF to all E.U. Member States and to the exporting countries [24,25,26]. In recent years, certain audits have been conducted in U.S. by the European Commission/Directorate-General for Health and Food Safety in response to continuing high number of RASFF notifications for AF contamination in tree nuts and peanuts imported from the U.S [1,27,28].

Despite tremendous efforts to control fungal toxin contamination of food and feed products, U.S. tree nut and peanut consignments destined to the E.U. countries are often being rejected due to AF levels exceeding not only the E.U. maximum limit but even the FDA action level. This serious matter prompted us to analyze RASFF notifications for the occurrence of mycotoxins in food and feed products of U.S. origin exported to the E.U. countries during the period of 2010 to 2019. We mostly focused on food samples because they accounted for 95.7% of all food and feed notifications, and we considered AF contamination as the target for this study as it represents 98.9% of mycotoxin notifications. In this report, we examined the levels of AF contamination in food products of U.S. origin, notification years, the types of rejection, the countries issuing the most notifications, and the product category that is frequently contaminated with AFs. We aim to provide ready-to-access data on mycotoxin contamination of food and feed products originating from the U.S. and exported to the E.U. countries. Moreover, the numbers of RASSS mycotoxin notifications issued for food products from other countries exported to E.U. countries were compared to those issued for U.S. food products. Along with other tracing tools, this study will help to explore the root problems contributing to AF contamination in U.S. tree nuts and may help to assess risk factors associated with the problem.

## 2. Results

### 2.1. RASFF Mycotoxin Notifications for U.S.-Originated Food Products and the Affected Food Category

Between 2010 and 2019, 442 mycotoxin notifications were reported for food products originated in the U.S. Of these, 98.9% (437) notifications were reported for AFs. Most of these notifications (97.7%) were associated with nuts (almonds, pistachios and peanuts). Of these, 22% (99) were reported for almonds, 42% (187) for pistachios, and 34% (151) for peanuts (Figure 1A). As reported by the RASFF portal, the products for which almond notifications occurred included 57% “almond” (*n* = 56), 20% “shelled almond” (*n* = 20), 8% “almond in shell” (*n* = 8), 2% “salted and roasted almond” (*n* = 2), 6% “almond kernels” (*n* = 6), 5% “whole almond with skin” (*n* = 5), and 1% for “peeled almond” (*n* = 1). Pistachio notifications were distributed as follows: 25% “pistachio in shell” (*n* = 67), 37.7% “pistachio nut” (*n* = 71), 7.9% “shelled pistachio” (*n* = 15), 7.4% “salted and roasted pistachio” (*n* = 12), 6.3% “pistachio kernels and chopped” (*n* = 12), and 4.7% “raw pistachio” (*n* = 9). For peanut (groundnut) notifications, notifications occurred in the following categories: 50% “peanut” (groundnut; *n* = 69), 22.7% “shelled peanut” (*n* = 31), 7.4% “blanched peanut” (*n* = 15), 7.3% “peanut kernels” (*n* = 10), 6.6% “peanut in shell” (*n* = 9), and 1.4% “salted and roasted peanut” (*n* = 2) (Figure 1B).

Very few notifications were reported by RASFF for apricot kernels (*n* = 1), peanut butter (*n* = 4), shelled walnuts (*n* = 1), and pecan nuts (*n* = 4) during 2010 to 2019. Only five notifications (1.13%) were reported for other mycotoxins in U.S. originated food during this time period. Two notifications were for the ochratoxin A contamination in ground corn for tortillas (37.32 ng g^−1^) and spaghetti (7200 ng g^−1^). Three notifications were reported for deoxynivalenol in wheat (*n* = 2; 1676 and 3138 ng g^−1^) and maize (*n* = 1; 2688 ng g^−1^).

### 2.2. Levels of AF Contamination in the U.S. Nuts as Specified in RASFF Notifications

The percentage of U.S. nuts containing AFs from 2010 to 2019 notified by RASFF is illustrated in Figure 2. We allocated the RASFF notifications by the levels of AF contamination in U.S. nuts into three groups: >4 to ≤10 ng g^−1^ as group one, >10 to <20 ng g^−1^ as group two, and ≥20 ng g^−1^ (FDA action level) as group three. Based on this classification, we found that 19% of notifications fell into group one (*n* = 81), 23% into group two (*n* = 98), and 57.9% of the notifications into group three (*n* = 247).

### 2.3. Notification Types and Notifying Countries

Figure 3A shows the numbers of the RASFF AF notifications based on the type of notifications for 2010 to 2019. The RASFF notifications for AF in the U.S. originated nuts were classified as border rejection (78%), alert (5.8%), and information for attention and follow-up (15.8%). The top five major notifying countries were the Netherlands with 27% of notifications (*n* = 126), Spain with 12.33% of notifications (*n* = 57), Italy and the United Kingdom with 11.4% of notifications each (*n* = 53 each), and Germany with 10.3% of notifications (*n* = 48). Other E.U. countries reported 27% of RASFF notifications (Figure 3B). In 2019, the top five major notifying countries were the Netherlands with 39% of notifications, the United Kingdom with 16% of notifications, Italy with 13% of notifications, Spain with 11% of notifications and Germany with 7.0% of notifications. Other E.U. countries reported 14% of RASFF notifications (Figure 3C).

### 2.4. RASFF Mycotoxin Notifications on the U.S. Originated Feed Products and the Affected Feed Category

Twenty mycotoxin notifications for feed products of U.S. origin were reported by RASFF during 2010 to 2019. All of these notifications were reported for AF contamination of groundnuts for birdfeed and wildlife. Levels of total AF contamination ranged from 28 ng g^−1^ to 220 ng g^−1^. Ten notifications reported total AF levels in the range of >20 and ≤50 ng g^−1^, seven notifications fell in the range of >50 and ≤100 ng g^−1^, and three notifications reported AFs levels of more than 100 ng g^−1^. About 80% of these notifications were classified as border rejections and the other 20% were classified as information for attention.

### 2.5. RASFF Mycotoxin Notifications for Global Food and Feed Products for 2010 to 2019

RASFF reported 5045 and 439 notifications for mycotoxin contamination in food and feed products, respectively, exported to E.U. countries from around the world during the years 2010 to 2019. Amongst food notifications, 89% (*n* = 4487) of notifications for mycotoxin contamination were attributed to AF contamination. The second most reported mycotoxin was ochratoxin A with 10% (*n* = 507) of the RASFF notifications. Deoxynivalenol, fumonisins, zearalenone and patulin were reported in 1.01% (*n* = 51), 0.71% (*n* = 36), 0.23% (*n* = 36) and 0.09% (*n* = 5) RASFF notifications, respectively (Figure 4).

The top 10 countries linked to 80% of RASFF mycotoxin notifications on food products were Turkey (32.7%), China (15.1%), India (12.2%), U.S. (10.7%), Iran (9.5%), Argentina (8.0%), Egypt (4.8%), Brazil (2.6%), Pakistan (1.7%), Nigeria (1.5%), and Ghana (1.3%) (Figure 5A). However, during 2010 to 2019, mycotoxin notifications were reported for more than 97 countries including the E.U. countries (Figure 5B).

Regarding feed products, AF contamination was reported in 98.6% (*n* = 433) of the RASFF notifications. Three notifications were reported for zearalenone, two for T-2 and HT-2 toxins and one notification was recorded for deoxynivalenol. The most frequently contaminated feed were groundnuts (*n* = 342), maize (*n* = 51), sunflower seeds (*n* = 16), cotton seeds (*n* = 4), rice bran (*n* = 4), sorghum (*n* = 3), compound feed (*n* = 3), and others (*n* = 16).

## 3. Discussion

### 3.1. Summary of the Analysis of the RASFF Notifications

To the best of our knowledge, there are no publications or available reports which assess RASFF notifications for AF contamination of the U.S. nuts. In this work, we primarily analyzed RASFF AF notifications for U.S. food products exported to E.U. countries during the last ten years (2010 to 2019). We found that 98.9% of the notifications were reported for AF contamination of nuts (almonds, peanuts, and pistachios). The most frequent notifications were reported for pistachios (42%) followed by peanuts (34%) and almonds (22%). All kinds of nuts such as shelled nuts, nuts in the shell, kernels nut, or roasted and salted nuts were reported to have AFs in the RASFF notifications, with different prevalences and levels of contamination. More than half of these notifications (57.9%) reported total AF levels greater than the U.S. FDA action level for food (20 ng g^−1^). About 19% of notifications reported AF levels in the range of >4 to ≤10 ng g^−1^, and 23% reported AF levels between >10 and ≤20 ng g^−1^. The Netherlands issued more of these notifications than any other E.U. country, with issuing more than 27% of AF notifications for U.S. nuts. The numbers of border rejections for U.S. nuts exported to E.U. countries, as cited in RASFF notifications, have been increased due to AF contaminations in the last few years. The number of such notifications was 87 in 2019, 96 in 2018, 52 and 2017, 44 in 2016, and 34 in 2015.

Border rejections constituted more than 78% of notifications for AF contamination in nuts that originated in the U.S. Only 1.13% of U.S. nuts were reported to be contaminated with other mycotoxins. Ochratoxin A contamination was reported in two notifications in ground corn and spaghetti. Deoxynivalenol was reported in three notifications associated with wheat and maize. No notifications were reported for patulin, fumonisins, zearalenone, and patulin.

We analyzed the RASFF notifications for mycotoxins in feed products of U.S. origin. We identified 20 relevant notifications. Interestingly, all these notifications reported AF contamination in groundnuts for birdfeed and wildlife. The levels of AF contaminations were 28 ng g^−1^ to 220 ng g^−1^. About 80% of these notifications were classified as border rejections, with the other 20% classified as information for attention. The country issuing the most notifications for mycotoxins in feed products of U.S. origin was the United Kingdom followed by the Netherlands.

For the years 2010 to 2019, RASFF reported 5045 and 439 notifications on mycotoxin contaminations in food and feed products, respectively, exported to E.U. countries from all countries around the world. The U.S. is the fourth top country linked to the notifications, behind Turkey, China, and India. Among the notifications for food products from all countries, 89% (*n* = 4487) of the reported notifications attributed to AF contamination. The second most reported mycotoxin in food products was ochratoxin A, which was responsible for 10% (*n* = 507) of the RASFF notifications. Deoxynivalenol, fumonisins, zearalenone, and patulin were reported in less than 2% of notifications.

Regarding the feed products from all countries, AF contamination was reported in 98.4% of the RASFF notifications. Of these, 77.9% of the contaminated feed products were groundnuts and 11.6% were maize. Sunflower seeds, cottonseeds, rice bran, sorghum, and compound feed were also reported to be contaminated with AFs.

### 3.2. Problem Characterization

Due to frequent RASFF notifications for U.S. nuts destined to E.U. countries in recent years, some assessments have been conducted by the European Commission/Directorate-General for Health and Food Safety in the U.S. for pistachios and peanuts [27,28]. Overall, highly limited official U.S. control measures to tackle AF contaminations were noted, according to the findings of the most recent audit on peanuts (7–15 October 2019). In addition, sampling for AFs and analysis and validation of data for peanuts intended for export to the E.U. has not been performed to meet the 100% of the requirements of Regulation (EC) No. 401/2006 [27]. Another audit on U.S. pistachios that was conducted 5–12 September 2017, found that there were no official controls or requirements applicable to pistachios intended for export to E.U. In addition, there were no legal requirements applicable to these exports to ensure those sampling methods, analyses or reporting procedures fulfilled Regulation (EC) No 401/2006. Furthermore, samples rejected by RASFF due to AF contamination were not adequately followed up in the U.S. system to investigate possible root causes or to implement preventive measures [28].

While the USDA has implemented an instrument called AF certification for peanuts and tree nuts and has also adopted pre-export controls and check on AFs in U.S. nuts, most of these programs are voluntary. Most companies exporting to E.U. have signed memorandum of understanding to comply with these programs. No official controls are performed to verify compliance with this program, and there is no official supervision of consignments destined for the E.U. relating to AF contamination [27,28].

Another possible explanation for the large number of RASFF notifications on U.S. nuts is the gap between the FDA action level and the E.U. maximum level of contamination. The U.S. FDA has established an action level of 20 ng g^−1^ for total AFs (B1, B2, G1 and G2) for foods, peanut, and peanut products, and pistachio nuts [1,29]. European maximum levels for AF contamination of groundnuts, tree nuts, and processed nut products for direct human consumption are 2.0 ng g^−1^ for AFB1 and 4.0 ng g^−1^ for total AFs. If the nuts are subject to sorting or other physical treatment before consumption or will be used as an ingredient in foodstuffs, the limits for AFB1 and total AFs are 8.0 ng g^−1^ and 15 ng g^−1^ for groundnuts and 5 ng g^−1^ and 10 ng g^−1^ for tree nuts, respectively [30].

The lack of a surveillance program and a regular monitoring system to detect AF contamination and levels of contamination represent significant drawbacks that the U.S. is facing in the fight against AFs. No studies for the occurrence and levels of AFs in the U.S. nuts for 2010~2019 have been published. Regular monitoring and testing are insufficiently employed at the level of individual states. The results of the states’ monitoring and surveillance packages could serve as a useful tool if the levels of AFs are elevated in a geographical area, alerting growers to the need for more attention and testing.

## 4. Conclusions and Recommendations

Almost 95% of U.S. mycotoxin RASFF notifications were reported for foods and only 5% for feeds. The number of E.U. RASFF notifications and border rejections of U.S. nuts, mainly pistachios, almonds, and peanuts, due to contamination with AFs have increased over last ten years (2010 to 2019). More than 50% of notifications were due to AF levels not only exceeding the E.U. maximum limits but also the U.S. FDA’s action level. Pistachios were the nut type responsible for the most notifications over the last ten years. Border rejections constituted more than 78% of RASFF notifications for AF contamination in U.S. nuts destined to E.U countries.

RASFF reported 5045 and 439 notifications for mycotoxin contamination in food and feed products, respectively, exported to E.U. countries from around the world during the years 2010 to 2019. About 89% of food and 98.6% of feed notifications for mycotoxin contamination were attributed to AF contamination.

The growing numbers of reports on the health benefits of eating nuts will likely lead to increased consumption of these products. In addition, the food industry using large numbers of tree nuts to manufacture pastries, sweets, ice cream, and confectionary products. As U.S. continues to be the largest supplier of tree nuts to the globe, especially to E.U. countries, it is therefore of great importance to keep U.S. nuts sheltered from AF contamination by implementing a mandatory and enforceable legal framework for official export control procedures concerning AFs in nuts.

To avoid escalating numbers of E.U. RASSF notifications, U.S. sampling, method validation, and results and reporting should comply with the E.U. requirements of Regulation (EC) No 401/2006. Importantly, comprehensive surveillance data on the occurrence and levels of AFs in almonds, pistachios, and peanuts are urgently needed to assess the current and ongoing conditions of the problem. The rejected shipments of the nuts exported from U.S. to E.U. countries due to AF contaminations should be adequately followed up on in order to identify possible root causes and/or to implement preventive measures.

## 5. Material and Methods

Data were obtained from the RASFF portal (https://webgate.ec.europa.eu/rasff-window/portal/?event=SearchForm&cleanSearch=1). Search criteria for the RASFF mycotoxins notifications in the U.S food and feed products over ten years (2010 to 2019) were as follows: Type “food and feed”, Hazard category “mycotoxins”, notified between “01/01/2010” and “31/12/2019”, product flagged as “origin”, product country “United States (US)”. Search criteria for the worldwide RASFF notifications in food products over ten years ((2010 to 2019) were: Type “food”, Hazard category “mycotoxins”, notified between “01/01/2010” and “31/12/2019”, product flagged as “origin”, product country “not specified”. The extracted data were exported from the RASFF portal directly to Microsoft Excel 2010 (Microsoft 365 MSO) to generate descriptive statistics. Each single notification list contained data in the following order: product category, date, reference, product type, notification type, notification basis, notified by, countries concerned, subject, action taken, distribution status, and risk decision. When more than one originating country was mentioned, or two countries (one for raw product origin and the other for processing and packaging), we considered the country of origin of the raw food.

Levels of AF contamination were extracted from the “subject” category into the Excel column. Usually, the RASFF portal presents the results on AFB1 and total AFs (B1, B2, G1, G2). In this report, we considered the total AFs because the FDA action level is set for total AFs. Therefore, AF concentrations in this report represent the summation of four AFs (B1, B2, G1, G2).

There are three major types of RASFF notifications: alert, information, and border rejection. Alert notifications typically are delivered through RASFF when the hazard is detected in food and feed that are already present in the E.U. market and a rapid action is required to protect the public. Information notifications are used when a hazard is detected in food or feed placed in the market of one E.U. country but has not reached other E.U. members’ markets. In this case, the risk does not require rapid actions. Regulation (E.U.) No 16/2011 defines two sub-types of information notifications: ‘information notifications for follow-up’ and “information notifications for attention”. Border rejections are concerned with food and feed consignments that have been rejected at the external borders of the E.U. due to the presence of hazard in food and feed [21,24,31].

## Figures and Tables

**Figure 1 toxins-13-00090-f001:**
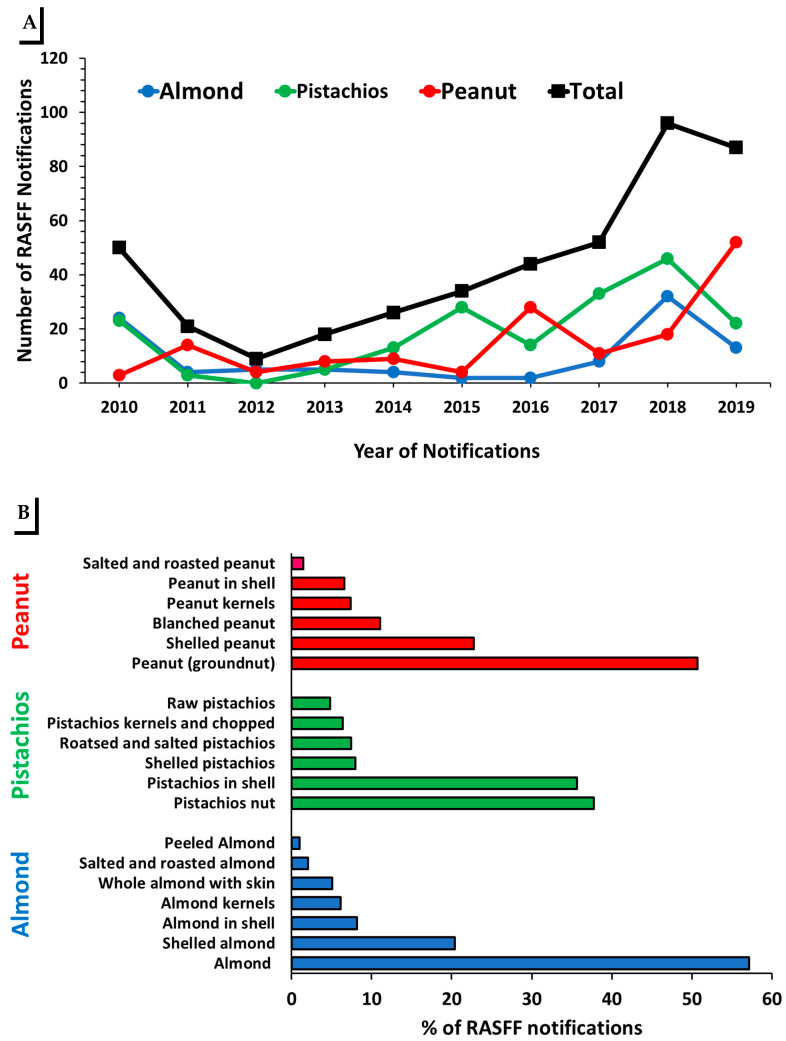
RASFF aflatoxin notifications in nuts of U.S. origin. (**A**) Number of RASFF notifications for aflatoxin in U.S. reported as food products during 2010–2019. Total number of notifications is shown in black. (**B**) Distribution of RASSF notifications for nuts of U.S. origin reported during 2010–2019 as food products. Almonds (blue), pistachios (green) and peanuts (red).

**Figure 2 toxins-13-00090-f002:**
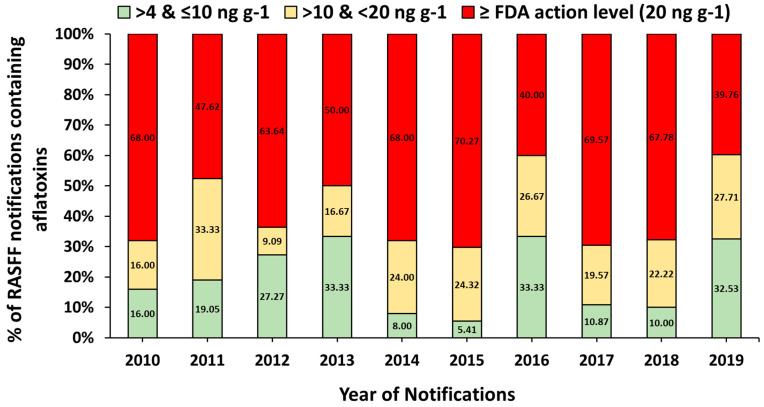
Percentage of U.S. nuts (as foods) containing aflatoxin for 2010 to 2019 as notified by RASFF.

**Figure 3 toxins-13-00090-f003:**
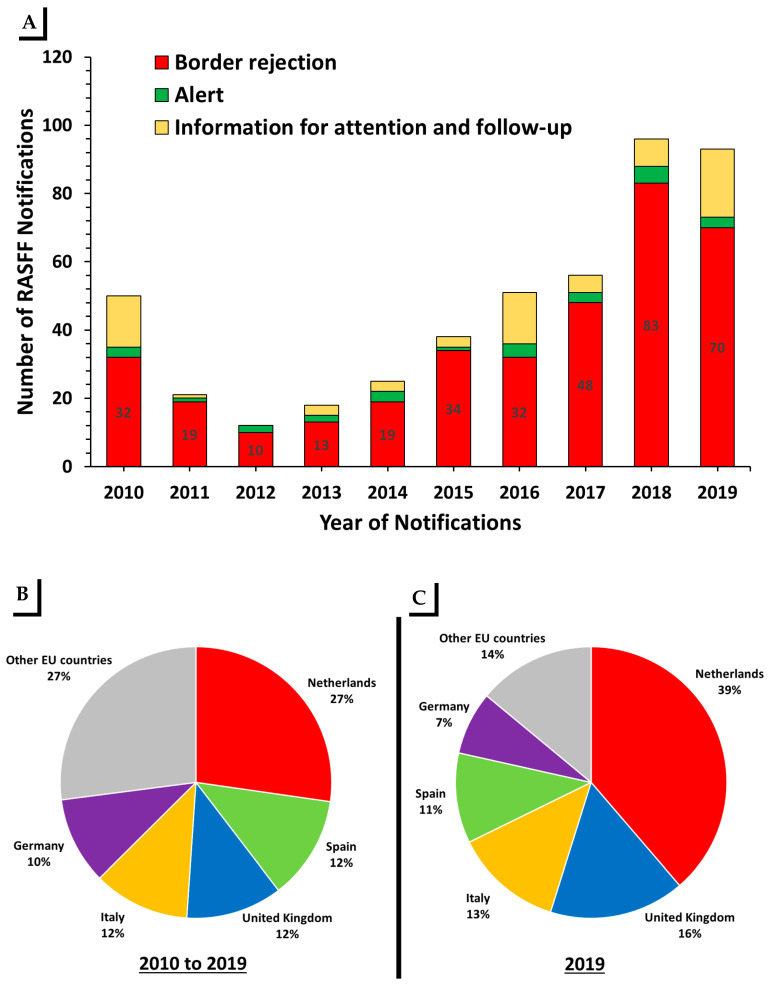
RASFF Notifications on U.S. food products and the top five notifying countries. (**A**) Number of RASFF notifications for aflatoxin contamination in food products based on type of notification for 2010 to 2019. (**B**,**C**) Percentage of RASFF notifications for aflatoxin in U.S. nuts (as food products) according to notifying country from 2010 to 2019 (**B**) and in the year 2019 (**C**).

**Figure 4 toxins-13-00090-f004:**
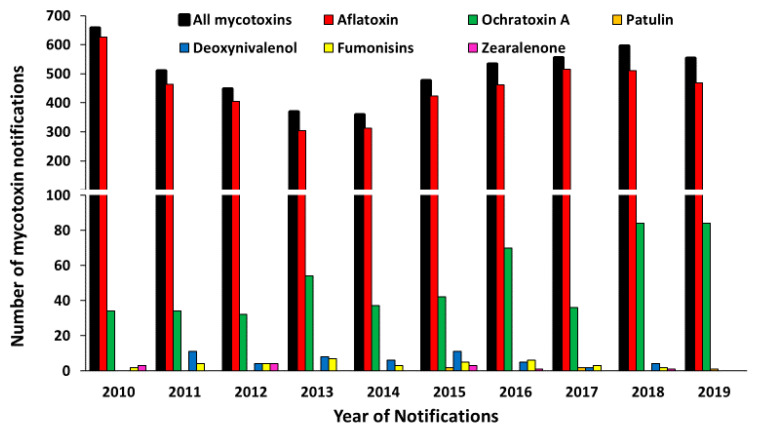
Total number of RASFF notifications for mycotoxins per year over ten years in food products from all countries over 2010 to 2019.

**Figure 5 toxins-13-00090-f005:**
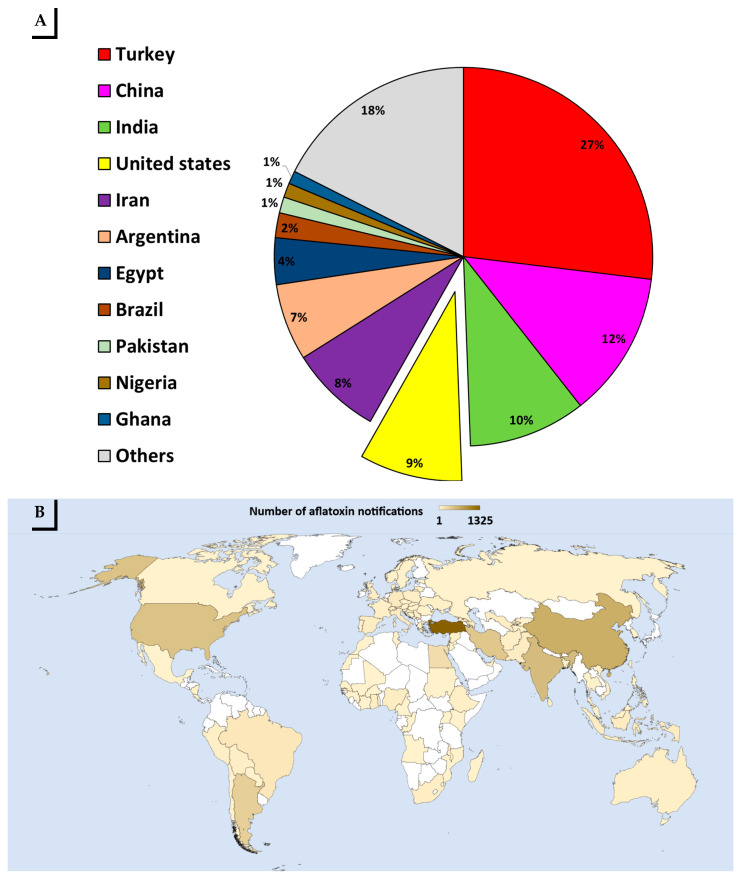
Source of RASFF notifications for mycotoxins in food products from all countries. (**A**) Percentage of RASFF notifications for mycotoxins in food products according to the country of origin. (**B**) Number of RASFF notifications for mycotoxins in food products according to the country of origin during 2010 to 2019.

**Table 1 toxins-13-00090-t001:** Marketing value of top 10 U.S. agricultural export for 2015–2019.

	Products	2015	2016	2017	2018	2019
1	Soybeans	18,862	22,839	21,456	17,063	18,660
**2**	**Tree Nuts**	**8441**	**7902**	**8479**	**8514**	**9075**
3	Beef and Beef Products	6311	6360	7263	8360	8090
4	Corn	8271	9879	9112	12,467	7617
5	Pork and Pork products	5567	5936	6485	6403	6952
6	Prepared Food	5849	6188	5938	6245	6682
7	Wheat	5628	5346	6058	5389	6214
8	Cotton	3902	3967	5845	6557	6153
9	Dairy Products	5240	4698	5377	5498	5931
10	Soybean Meal	4781	4073	3881	5079	4405
	All Others	60,206	57,558	58,266	58,021	56,880
	Total exported	135,073	136,762	140,177	141,614	138,678

Values in millions of dollars. Ethanol is not counted as an agricultural good in the USDA definition of agriculture. Source: United States Agricultural Export Yearbook, USDA, 2019.

**Table 2 toxins-13-00090-t002:** Values of top 10 export markets for U.S. Tree Nuts for 2015–2019.

	Products	2015	2016	2017	2018	2019
**1**	**European Union**	**2977**	**2585**	**2707**	**2769**	**3114**
2	India	606	521	738	663	823
3	Canada	686	598	643	696	696
4	Hong Kong	846	1156	1251	1052	692
5	China	208	182	243	328	606
6	United Arab Emirates	430	310	301	304	439
7	Japan	480	374	398	433	416
8	Mexico	269	253	256	371	344
9	Turkey	300	365	308	279	341
10	South Korea	354	296	305	290	290
	All Others	1285	1261	1329	1331	1314
	Total Exported	8441	7901	8479	8516	9075

Values in millions of dollars. Source: United States Agricultural Export Yearbook, USDA, 2019.

## Data Availability

The data presented in this study are openly available in the European Commission RASFF portal database (https://webgate.ec.europa.eu/rasff-window/portal).

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
