# Peer review of "Analysis of E.U. Rapid Alert System (RASFF) Notifications for Aflatoxins in Exported U.S. Food and Feed Products for 2010–2019"

_toxins, 2021, doi:10.3390/toxins13020090_

Round 1
Reviewer 1 Report
This article is an interesting compilation of the data offered by RASFF on nut exports and their rejection or acceptance based on the presence of miscotoxins in them.
In this data collection the authors should add the general criteria established by RASFF for the rejection of such products in order to better understand how the process is carried out.
The authors should make a more in-depth assessment of the results and consequences of these, otherwise the article becomes a web-based data collection.
The authors should draw more concrete conclusions before the article can be published.
Author Response
Comment. We thank the reviewers for the constructive criticism.
Authors response:
Reviewer 1:
This article is an interesting compilation of the data offered by RASFF on nut exports and their rejection or acceptance based on the presence of miscotoxins in them.
Answer: Thank you.
In this data collection the authors should add the general criteria established by RASFF for the rejection of such products in order to better understand how the process is carried out.
Answer: in the line 241 – 247, we clearly mentioned the maximum allowable limit for aflatoxin as established in EU countries and FDA as well. U.S. food and feed shipments have a total aflatoxin more than EU limit, after being tested by reference laboratory, would be rejected and the whole shipment is not allowed to enter the EU border. We clearly indicated that in material and method part, line 303-305.
The authors should make a more in-depth assessment of the results and consequences of these, otherwise the article becomes a web-based data collection.
Answer: the authors are agreeing with the reviewer; unfortunately, the authors are unable to access more details specifically about the size of shipment, the fate of rejected shipments and even the cost and burden of that on the traders and exporters. The main objective of this study is to shed the light on the occurrence of aflatoxin and other mycotoxin of food and feed nuts that destinated to E.U. countries. It provides a summary and easy to access data to anyone who is not familiar with RASFF portal.
The authors should draw more concrete conclusions before the article can be published.
Answer: we draw a simple and direct message in the conclusion part about the main findings and we proposed some measures that may help to minimize the risk of shipment rejection by the E.U border due to aflatoxin contamination.
Reviewer 2 Report
This manuscript investigates RASFF notifications to the United States for food and feed originating from the US for a duration of ten years. The authors bring out some alarming results of AF levels in US nuts despite of the stringent regulatory levels set up by the FDA in US.
Some minor corrections attached in the pdf can improve the manuscript. A major negative aspect of this manuscript is that the authors have not referenced several sentences that require citations. I have marked all of these. Also, please include relevant research article citations in addition to citing review articles.

Author Response
Comment. We thank the reviewers for the constructive criticism.
This manuscript investigates RASFF notifications to the United States for food and feed originating from the US for a duration of ten years. The authors bring out some alarming results of AF levels in US nuts despite of the stringent regulatory levels set up by the FDA in US.
Answer: Thank you! We see this paper as alarming document that draw serious concern not only for U.S. international trade but also for those nuts marketed domestically.
Some minor corrections attached in the pdf can improve the manuscript. A major negative aspect of this manuscript is that the authors have not referenced several sentences that require citations. I have marked all of these. Also, please include relevant research article citations in addition to citing review articles.
Answer: we corrected all minor issues. We added more relevant citations that you have highlighted. We added research article in addition to cite some valuable review paper.
Minor revision:
Abstract exceeds the 200 word limit. Please rectify as per journal requirements.
Answer: We make it 252 words. We hope the editor accepts that as it more coherent and attractive in this length. However, we are ready to minimize it further.
Lines 31 to 33 need relevant citations for contamination of mentioned crops. Please include relevant research article citations in addition to review articles for convenient access to original work. Extensive amount of work has been published on contamination of cereals, tree nuts and spices in the US itself.
Answer: We added three research paper references (5, 6 and 7).
Line 36- Remove the word “plants”.
Answer: removed
Lines 35 to 37 require appropriate citations.
Answer: 2 references were added (8,9).
Lines 40 to 43 require appropriate citations.
Answer: References 14-17 are basically covered this.
Line 66 and 259- FDA action level (20 ppb) is less conservative than EU maximum limits (2, 4 or 10 ppb depending on the commodity). Please rewrite these sentences. It is currently being interpreted as that US action levels are more conservative than that of EU, which is not true.
Answer: we removed “conservative”. Despite tremendous efforts to control fungal toxin contamination of food and feed products, U.S. tree nut and peanut consignments destined to the E.U. countries are often being rejected due to AF levels exceeding not only the E.U. maximum limit but even the FDA action level.
Line 76- RASSF is missing an F. Lines 82, 111, 138- U of U.S. should be in upper case.
Answer: we corrected the missing letter, correct is RASFF. We have U.S. as upper case but the font make it look like lower case.
Line 136- Please change to “..notifying country from 2010 to 2019”
Answer: we replaced for by from
Line 141- Please change to “All of these notifications….” Line 209- Please remove the extra space before “Among the notifications…”. Lines 233 to 238 need relevant citations. Line 292- Please replace “presented” by “presents”. Line 293- “…FDA action level is set…”
Answer: we added “All of these notifications”. We deleted the space. We added 2 references. Presented replaced by presents. Was replaced by is.
Reviewer 3 Report
The reviewed manuscript, entitled ‘Analysis of E.U. Rapid Alert System (RASFF) Notifications For Aflatoxins in Exported U.S. Food and Feed Products for 2010- 2019’ summarizes in a very comprehensive and clearly arranged way the numbers of toxin contaminations mainly in nuts in E.U. imported food and feed. Up to now, there is no paper describing this specific field and giving an easy understandable overview about this topic.
Within introduction, the authors give sufficient background information. However, some statements, especially in line 42/43 describing the market volume of the U.S. tree nut and peanut exports, are not sufficiently documented. The given information needs to be backed up with a reference. Additionally, a statement explaining the testing volume for mycotoxins might further depict the importance of mycotoxin testing. This could explain a possible difference in deviating test frequency of the different mycotoxins and consequently provide additional support for the rationale of selecting aflatoxin as most prominent mycotoxin, as stated in line 70.
In chapter 2, there are several points which have to be improved. First, the authors should think about reducing the amount of written information (line 88-98). Giving examples and just referring to respective figure might be possible, thus enhancing reading-efficiency. For further enhancement of reading-flow, the authors should check their manuscript for consistency. Here, especially in line 84-98, the order of nuts should be the same (almonds, pistachios, peanuts), as written in the text 2.1 and shown in Figure 1A. Please switch the order of nuts in 1B.
The authors should add information on the overall U.S. based E.U. imports of respective products. This can e.g. be achieved by adding a second axis in figure 1a and presenting the total imports of respective food products over the years. Based on such export/import statistics, it should be analyzed and discussed, if the export/import level of the different nuts changed over the years and in how far this might also explain the increase of notifications/border rejections?
Regarding Figure 1, a larger gap between panel A and B should be inserted. Currently, title of X-axis corresponding to Figure 1A looks like the heading of panel 1B. A further point which has to be optimized is the visual interpretation of data in Figure 1B. With the visualization of data in combination with the X-axis title, it seems as about 50% of the RASFF notifications would be for peanut (groundnut), about 55% for almond, and 35% for pistachio (nut), which in total would be more than 100% of all RASFF notifications. Accordingly, it must be made clear that of the respective nut as a total quantity e.g. 50% are for peanut (groundnut). A much better representation which directly state the most heavily contaminated nut, would be a recalculation of the amount of notifications per nut related to the entire quantity of RASFF notifications. This means a total of 437 RASFF notification of AFs, of those 56 notifications for almond, which would be in total 12.8%. If not changing graph, a renaming of X-axis title has to be done in e.g. “% RASFF Notifications for respective nut in 2010-2019” or any else.
Line 105-110 is somewhat incoherent, since chapter 2.1 only deals with the numbers of the contaminations and not with the levels. This paragraph would rather fit into chapter 2.2. Simply copying down the text block would be sufficient.
In the following chapters, the color scheme has to be adapted for keeping the scheme as consistent as possible (e.g. green of group one in Figure 2 and green of Alert in Figure 3A). Additionally try to keep the same format. Possibly, change the order, so that border rejections (Figure 3A) is on top of the bar since the largest group of Figure 2 is also on top of bar. This would enhance the reading-/understanding-efficiency for your readers since largest group is also on top of bar in Figure 2.
Figure 3 – Keep panel descriptions A), B), C) always on the left-hand side of each panel. Check Fig 3 B) and C). It is not clear why the authors show year 2019 (Figure 3C) separately. It should be mentioned/discussed that there is a high probability of a correlation between notifying country and point of entry for the exported nuts to the E.U. Hence, it is not surprising that the Netherlands reported most notifications since Rotterdam is the largest E.U. seaport and most imports reach the E.U. via this seaport/the Netherlands, especially from the U.S. The authors might also check and mention similar correlations to other notifying countries.
Please provide a more appropriate introduction in chapter 2.4. as there is a very strong change of topic that interrupts the flow of reading.
Chapter 2.5 – Especially for better visualization, a split of Y-axis at e.g. 100 number of mycotoxin notifications of graph in Figure 4A would be very helpful, thus resulting in better depiction of low numbers. Additionally, a second Y-axis would be unnecessary if the title of the left Y-axis would be changed from ‘Number of individual mycotoxin notifications’ to ‘Number of mycotoxin notifications’. In addition, it should be checked if panel 4B does not represent too much redundancy to panel 4A, which after the above optimization clarifies the information in a more scientific manner. My suggestion would be a deletion of panel 4B, just focusing on panel 4A. Please check if figure description of Fig. 4 is actually adequate for describing the content.
Please check for better contrast in Figure 5B and, if necessary, cite the source of this figure.
Line 170-171, please sort from high to low, as ‘Three notifications were recorded for zearalenone, two for T-2 and HT-2 toxins, and one for deoxynivalenol’. This also would enhance the reading efficiency.
In Section 3 and 4, the authors discuss their analyzed data and speculate about possible reasons in an appropriate way. Nonetheless, especially in line 250, a more scientific way of writing should be used. Obviously, aflatoxins are highly carcinogenic and have harsh effects on human and animal, but please think about wording as ‘silent killer’ (line 250).
Please check for spelling mistakes as line 76, 268 and further lines in your manuscript. Make sure ng g– 1 is written correctly. Make sure, number and whole unit are in one line (line 142) within manuscript. Please add the respective number of 97.7% in line 86. Line 125 – please check consisting typing of %-numbers. Check e.g line 124 and 125, here the numbers 15.8%, 27% as well as 12.33% are shown. Please check consistency as mentioned above (almonds, pistachios, peanuts), line 86, Fig. 1B, line 257 and further lines.
Line 19: Change ‘All U.S. Feed notifications…’ to ‘All U.S. feed notifications…’
Line 21: Change ‘E.U’ to ‘E.U.’
After a careful revision and implementation of the above mentioned notes, the manuscript will support the research and the manuscript will be worth to publish.
Author Response
Comment. We thank the reviewers for the constructive criticism.
The reviewed manuscript, entitled ‘Analysis of E.U. Rapid Alert System (RASFF) Notifications For Aflatoxins in Exported U.S. Food and Feed Products for 2010- 2019’ summarizes in a very comprehensive and clearly arranged way the numbers of toxin contaminations mainly in nuts in E.U. imported food and feed. Up to now, there is no paper describing this specific field and giving an easy understandable overview about this topic.
Answer: thank you!
Within introduction, the authors give sufficient background information. However, some statements, especially in line 42/43 describing the market volume of the U.S. tree nut and peanut exports, are not sufficiently documented. The given information needs to be backed up with a reference. Additionally, a statement explaining the testing volume for mycotoxins might further depict the importance of mycotoxin testing. This could explain a possible difference in deviating test frequency of the different mycotoxins and consequently provide additional support for the rationale of selecting aflatoxin as most prominent mycotoxin, as stated in line 70.
Answer: we cited 4 references for the whole paragraph as it is connected (Ref 10-13, now 14-17). To your knowledge, data was taken directly from the USDA, Foreign Agriculture Service.
Testing volume: Aflatoxin is the most reported mycotoxins not only in U.S. shipments but in whole world shipment destined to E.U. countries. Most of nuts, grains and other crops-based shipments are screened, within single run, for several mycotoxins including aflatoxins.
In chapter 2, there are several points which have to be improved. First, the authors should think about reducing the amount of written information (line 88-98). Giving examples and just referring to respective figure might be possible, thus enhancing reading-efficiency. For further enhancement of reading-flow, the authors should check their manuscript for consistency. Here, especially in line 84-98, the order of nuts should be the same (almonds, pistachios, peanuts), as written in the text 2.1 and shown in Figure 1A. Please switch the order of nuts in 1B.
Answer: we believe that breaking out the nuts into different category along with percentage and number of notifications is important to emphasize that aflatoxin contamination can occurred even with those roasted and salted. We changed the order in the text and in Fig 1B to be (almonds, pistachios, peanuts).
The authors should add information on the overall U.S. based E.U. imports of respective products. This can e.g. be achieved by adding a second axis in figure 1a and presenting the total imports of respective food products over the years. Based on such export/import statistics, it should be analyzed and discussed, if the export/import level of the different nuts changed over the years and in how far this might also explain the increase of notifications/border rejections?
Answer: unfortunately, we are unable to obtain such data for pistachios, peanuts and almonds over the entire study period (2010-2019) either as export size (metric tons) or as marketing values (millions of dollars). Further, there is substantial differences with different resources. However, we added 2 tables in the introduction part that emphasize and importance and marketing value of tree nut (2015-2019) : Table 1 Marketing value of top 10 U.S. agricultural export and table 2: Values of top 10 export markets for U.S. Tree Nuts. Table 1 is clearly indicated the importance of tree nuts as major U.S. agricultural export product. Table 2 can give clue about the export marketing values of tree nut especially for E.U. countries over last 5 years (2015 – 2019).
Regarding Figure 1, a larger gap between panel A and B should be inserted. Currently, title of X-axis corresponding to Figure 1A looks like the heading of panel 1B. A further point which has to be optimized is the visual interpretation of data in Figure 1B. With the visualization of data in combination with the X-axis title, it seems as about 50% of the RASFF notifications would be for peanut (groundnut), about 55% for almond, and 35% for pistachio (nut), which in total would be more than 100% of all RASFF notifications. Accordingly, it must be made clear that of the respective nut as a total quantity e.g. 50% are for peanut (groundnut). A much better representation which directly state the most heavily contaminated nut, would be a recalculation of the amount of notifications per nut related to the entire quantity of RASFF notifications. This means a total of 437 RASFF notification of AFs, of those 56 notifications for almond, which would be in total 12.8%. If not changing graph, a renaming of X-axis title has to be done in e.g. “% RASFF Notifications for respective nut in 2010-2019” or any else.
Answer: we inserted a gap between panel A and B. Figure 1 B showing the percentage of RASFF notification for each nut. We breakdown the notification within each nut. For example, almonds breakdown products: 57% “almond” (n=56), 20% “shelled almond”, 8% “almond in shell”, 2% “salted and roasted almond”, 6% “almond kernels”, 5% “whole almond with skin”, and 1% for “peeled almond”. The total would be 100%. Then we breakdown the notifications for peanut and pistachios. In Y axis, we clearly labeled that with the nut name.
Line 105-110 is somewhat incoherent, since chapter 2.1 only deals with the numbers of the contaminations and not with the levels. This paragraph would rather fit into chapter 2.2. Simply copying down the text block would be sufficient.
Answer: line 105-107 is relevant. Line 108-110 is concerning with mycotoxin notifications and we believe that its cant be fit under 2.2 as the later handling only aflatoxin notifications.
In the following chapters, the color scheme has to be adapted for keeping the scheme as consistent as possible (e.g. green of group one in Figure 2 and green of Alert in Figure 3A). Additionally try to keep the same format. Possibly, change the order, so that border rejections (Figure 3A) is on top of the bar since the largest group of Figure 2 is also on top of bar. This would enhance the reading-/understanding-efficiency for your readers since largest group is also on top of bar in Figure 2.
Answer: Figure 2 (aflatoxin concentration) and 3A (type of alert) is totally different and the color here does not reflect any similarity between them.
Figure 3 – Keep panel descriptions A), B), C) always on the left-hand side of each panel. Check Fig 3 B) and C). It is not clear why the authors show year 2019 (Figure 3C) separately. It should be mentioned/discussed that there is a high probability of a correlation between notifying country and point of entry for the exported nuts to the E.U. Hence, it is not surprising that the Netherlands reported most notifications since Rotterdam is the largest E.U. seaport and most imports reach the E.U. via this seaport/the Netherlands, especially from the U.S. The authors might also check and mention similar correlations to other notifying countries.
Answer: we moved the panel description to the left side. We need, in figure 3 C to provide the most recent data about the notifying countries in addition to what been mentioned in figure 3 B (overall). However, not all year the Netherlands was the notifying country, in 2010, 2016 and 2018, the Netherlands was not the # notifying country (data not shown).
Please provide a more appropriate introduction in chapter 2.4. as there is a very strong change of topic that interrupts the flow of reading.
Answer: We totally understand your concern, but we believe the current presentation is simple and easy to track by readers.
Chapter 2.5 – Especially for better visualization, a split of Y-axis at e.g. 100 number of mycotoxin notifications of graph in Figure 4A would be very helpful, thus resulting in better depiction of low numbers. Additionally, a second Y-axis would be unnecessary if the title of the left Y-axis would be changed from ‘Number of individual mycotoxin notifications’ to ‘Number of mycotoxin notifications’. In addition, it should be checked if panel 4B does not represent too much redundancy to panel 4A, which after the above optimization clarifies the information in a more scientific manner. My suggestion would be a deletion of panel 4B, just focusing on panel 4A. Please check if figure description of Fig. 4 is actually adequate for describing the content.
Answer: thank you so much for this great comment and guidance! We split Y axis, removed second access, and changed Y axis title. We deleted Figure 4 B.
Please check for better contrast in Figure 5B and, if necessary, cite the source of this figure.
Answer: we generated of this figure using excel based of RASFF analysis data. We improved the contrast.
Line 170-171, please sort from high to low, as ‘Three notifications were recorded for zearalenone, two for T-2 and HT-2 toxins, and one for deoxynivalenol’. This also would enhance the reading efficiency.
Answer: we sorted them.
In Section 3 and 4, the authors discuss their analyzed data and speculate about possible reasons in an appropriate way. Nonetheless, especially in line 250, a more scientific way of writing should be used. Obviously, aflatoxins are highly carcinogenic and have harsh effects on human and animal, but please think about wording as ‘silent killer’ (line 250).
Answer: we deleted “silent killer”.
Please check for spelling mistakes as line 76, 268 and further lines in your manuscript. Make sure ng g– 1 is written correctly. Make sure, number and whole unit are in one line (line 142) within manuscript. Please add the respective number of 97.7% in line 86. Line 125 – please check consisting typing of %-numbers. Check e.g line 124 and 125, here the numbers 15.8%, 27% as well as 12.33% are shown. Please check consistency as mentioned above (almonds, pistachios, peanuts), line 86, Fig. 1B, line 257 and further lines.
Answer: we corrected the mistakes.
Line 19: Change ‘All U.S. Feed notifications…’ to ‘All U.S. feed notifications…’
Answer: corrected. thanks
After a careful revision and implementation of the above mentioned notes, the manuscript will support the research and the manuscript will be worth to publish.
Reviewer 4 Report
In the manuscript presented to me for review, there is too little cited literature. The introduction is literary, but not enough detail. Too little literature cited. I have the impression that the article has a low scientific rating. The results describe the data that can be seen in the charts in too much detail. However, Figure 5B is not properly described in the text. Too few items of literature were used in the discussion that, contrary to what the authors say, is available.
Author Response
Comment. We thank the reviewers for the constructive criticism.
In the manuscript presented to me for review, there is too little cited literature. The introduction is literary, but not enough detail. Too little literature cited. I have the impression that the article has a low scientific rating. The results describe the data that can be seen in the charts in too much detail. However, Figure 5B is not properly described in the text. Too few items of literature were used in the discussion that, contrary to what the authors say, is available.
Response: we have added many relevant citations throughout the manuscript. Many of figures have been improved. The main purpose of this article is to shed the light and provide easy to access data on the occurrence of aflatoxin and other mycotoxin in U.S. shipment destined to E.U. To our knowledge, this is the first article that touch this important issue.
Round 2
Reviewer 4 Report
The authors did not add many relevant literature items to the manuscript, but only a few.
The authors should draw more concrete conclusions before the article can be published.
Author Response
The authors did not add many relevant literature items to the manuscript, but only a few.
The authors should draw more concrete conclusions before the article can be published.
Answer: Thank you!
We have modified the conclusion to include more significant data about RASFF notification of U.S. and worldwide shipments destined to E.U. during 2010-2019.
We have added adequate references and evidence (31 references) in a suitable and relevant spot throughout the entire article and we do not believe that further literature is necessary.